# Neural Networks Learning and Memorization with (almost) no Over-Parameterization

**Amit Daniely**
The Hebrew University and Google Research Tel-Aviv
amit.daniely@mail.huji.ac.il

## Abstract

Many results in recent years established polynomial time learnability of various models via neural networks algorithms (e.g. Andoni et al. [2014], Daniely et al. [2016], Daniely [2017], Cao and Gu [2019], Ji and Telgarsky [2019], Zou and Gu [2019], Ma et al. [2019], Du et al. [2018a], Arora et al. [2019], Song and Yang [2019], Oymak and Soltanolkotabi [2019a], Ge et al. [2019], Brutzkus et al. [2018]). However, unless the model is linearly separable Brutzkus et al. [2018], or the activation is quadratic Ge et al. [2019], these results require very large networks – much more than what is needed for the mere existence of a good predictor.

In this paper we make a step towards learnability results with near optimal network size. We give a tight analysis on the rate in which the Neural Tangent KernelJacot et al. [2018], a fundamental tool in the analysis of SGD on networks, converges to its expectations. This results enable us to prove that SGD on depth two neural networks, starting from a (non standard) variant of Xavier initialization Glorot and Bengio [2010] can memorize samples, learn polynomials with bounded weights, and learn certain kernel spaces, with *near optimal* network size, sample complexity, and runtime. In particular, we show that SGD on depth two network with $\tilde{O}\left(\frac{m}{d}\right)$ hidden neurons (and hence $\tilde{O}(m)$ parameters) can memorize $m$ random labeled points in $\mathbb{S}^{d-1}$.

## 1 Introduction

Understanding the models (i.e. pairs $(\mathcal{D}, f^*)$ of input distribution $\mathcal{D}$ and target function $f^*$) on which neural networks algorithms guaranteed to learn a good predictor is at the heart of deep learning theory today. In recent years, there has been an impressive progress in this direction. It is now known that neural networks algorithms can learn, in polynomial time, linear models, certain kernel spaces, polynomials, and memorization models (e.g. Andoni et al. [2014], Daniely et al. [2016], Daniely [2017], Cao and Gu [2019], Ji and Telgarsky [2019], Zou and Gu [2019], Ma et al. [2019], Du et al. [2018a], Arora et al. [2019], Song and Yang [2019], Oymak and Soltanolkotabi [2019a], Ge et al. [2019], Brutzkus et al. [2018]).

Yet, while such models has been shown to be learnable in polynomial time and polynomial sized networks, the required size (i.e., number of parameteres) of the networks is still very large, unless the model is linear separable Brutzkus et al. [2018], or the activation is quadratic Ge et al. [2019]. This means that the proofs are valid for networks whose size is significantly larger then the minimal size of the network that implements a good predictor[1].

In this paper we make a progress in this direction. We first consider the neural tangent kernel Jacot et al. [2018], which is a linearization of the functions that can be computed by the network, with weights that are close to a given weight vector $\mathbf{w}$. The NTK is one of the main technical tools in recent analysis of SGD on neural networks. Our first result is a near optimal bound on the rate in which the NTK converge to its expectation. We then utilize this results, and prove that it implies that SGD on depth two networks, starting form a (somewhat non-standard) variant of Xavier initialization Glorot and Bengio [2010] can learn memorization models, polynomials, and kernel spaces, with *near optimal* network size, sample complexity, and runtime (i.e. SGD iterations).

To the best of our knowledge, this is the first result which shows near optimal learnability of these models, and we believe that the result about NTK will be an essential tool for further progress, and in particular for proving a similar results for additional settings, architectures, and initialization schemes (particularly, the standard Xavier initialization). We next give more details about our results.

**Neural Network Algorithm**  We assume that the instance space is $\mathbb{S}^{d-1}$ and consider depth 2 networks with $2q$ hidden neurons. Such networks calculate a function of the form

$$h_{W,\mathbf{u}}(\mathbf{x}) = \sum_{i=1}^{2q} u_i \sigma\left(\langle \mathbf{w}_i, \mathbf{x} \rangle\right) = \langle \mathbf{u}, \sigma\left(W\mathbf{x}\right) \rangle$$

We assume that the network is trained via SGD, starting from random weights that are sampled from the following variant of Xavier initialization Glorot and Bengio [2010]: $W$ will be initialized to be a duplication $W = \begin{bmatrix} W' \\ W' \end{bmatrix}$ of a matrix $W'$ of standard Gaussians and $\mathbf{u}$ will be a duplication of the all-$B$ vector in dimension $q$, for some $B > 0$, with its negation. We will use rather large $B$, that will depend on the model that we want to learn.

**Bounded distributions**  Some of our results will depend on what we call the boundedness of the data distribution. We say that a distribution $\mathcal{D}$ on $\mathbb{S}^{d-1}$ is $R$-bounded if for every $\mathbf{u} \in \mathbb{S}^{d-1}$, $\mathbb{E}_{\mathbf{x}\sim\mathcal{D}}\langle \mathbf{u}, \mathbf{x} \rangle^2 \leq \frac{R^2}{d}$. To help the reader to calibrate our results, we first note that by Cauchy-Schwartz, any distribution $\mathcal{D}$ is $\sqrt{d}$-bounded, and this bound is tight in the cases that $\mathcal{D}$ is supported on a single point. Despite that, many distributions of interest are $O(1)$-bounded or even $(1 + o(1))$-bounded. This includes the uniform distribution on $\mathbb{S}^{d-1}$, the uniform distribution on the discrete cube $\left\{\pm\frac{1}{\sqrt{d}}\right\}^d$, the uniform distribution on $\Omega(d)$ random points, and more (see section A.5). For simplicity, we will phrase our results in the introduction for $O(1)$-bounded distribution. We note that if the distribution is $R$-bounded (rather than $O(1)$-bounded), our results suffer a multiplicative factor of $R^2$ in the number of parameters, and remains the same in the runtime (SGD steps).

**NTK Convergence**  For weights $(W, \mathbf{u})$ and $\mathbf{x} \in \mathbb{S}^{d-1}$ we denote by $\Psi_{W,\mathbf{u}}(\mathbf{x}) \in \mathbb{R}^{2q \times d}$ the gradient, w.r.t. the hidden weights $W$, of $h_{W,\mathbf{u}}(\mathbf{x})$. (A slight variant of) The NTK at $W$ is

$$k_W(\mathbf{x}, \mathbf{y}) = \frac{\langle \Psi_{W,\mathbf{u}}(\mathbf{x}), \Psi_{W,\mathbf{u}}(\mathbf{y}) \rangle}{2qB^2}$$

And the expected initial NTK is $k(\mathbf{x}, \mathbf{y}) = \mathbb{E}_W k_W(\mathbf{x}, \mathbf{y})$ Our main technical contribution is near optimal analysis of the rate (it terms of the size of the network) in which $k_W$ converges to $k$. Specifically, we show that for any $O(1)$-bounded distribution, and every function $f : \mathbb{R}^d \to \mathbb{R}$ in the kernel space $\mathcal{H}_k$ corresponding to $k$, there is a function $\hat{f}$ in the kernel space $\mathcal{H}_{k_W}$ corresponding to $k_W$ such that

$$\mathbb{E}_{\mathbf{x}\sim\mathcal{D}}(f(x) - \hat{f}(x))^2 = O\left(\frac{\|f\|_k^2}{dq}\right)$$

Here, $\|\cdot\|_k$ denotes the kernel norm of $f$. The proof of the aforementioned result is based on a new analysis of *vector* random feature schemes. While standard analysis of random feature schemes would lead to a bound of the form $\mathbb{E}_{\mathbf{x}\sim\mathcal{D}}(f(x) - \hat{f}(x))^2 = O\left(\frac{\|f\|_k^2}{q}\right)$, our new analysis show that for $O(1)$-bounded distributions, a factor of the input dimension $d$ can be saved.

As mentioned above, we utilize our result for NTK convergence to prove various learnability results for SGD on depth two networks.

**Memorization**    In the problem of memorization, we consider SGD training on top of a sample $S = \{(\mathbf{x}_1, y_1), \ldots, (\mathbf{x}_m, y_m)\}$. The goal is to understand how large the networks should be, and (to somewhat leaser extent) how many SGD steps are needed in order to memorize $1 - \epsilon$ fraction of the examples, where an example is considered memorized if $y_i h(\mathbf{x}_i) > 0$ for the output function $h$. Many results assumes that the points are random or "look like random" in some sense.

In order to memorize even just slightly more that half of the $m$ examples we need a network with at least $m$ parameters (up to poly-log factors). However, unless $m \leq d$ (in which case the points are linearly separable), best know results require much more than $m$ parameters, and the current state of the art results Song and Yang [2019], Oymak and Soltanolkotabi [2019a] require $m^2$ parameters. We show that if the points are sampled uniformly at random from $\mathbb{S}^{d-1}$, and the labels are random, then *any fraction* of the examples can be memorized by a network with $\tilde{O}(m)$ parameters, and $\tilde{O}\left(\frac{m}{\epsilon^2}\right)$ SGD iterations. Our result is valid for the hinge loss, and most popular activation functions, including the ReLU.

**Learning Polynomials**    For the sake of clarity, we will describe our result for learning even polynomials, with ReLU networks, and the loss being the logistic loss or the hinge loss. Fix a constant integer $c > 0$ and consider the class of even polynomials of degree $\leq c$ and coefficient vector norm at most $M$. Namely, $\mathcal{P}_c^M = \left\{ p(\mathbf{x}) = \sum_{|\alpha| \text{ is even and } \leq c} a_\alpha \mathbf{x}^\alpha : \sum_{|\alpha| \text{ is even and } \leq c} a_\alpha^2 \leq M^2 \right\}$ where for $\alpha \in \{0, 1, 2, \ldots\}^d$ and $\mathbf{x} \in \mathbb{R}^d$ we denote $\mathbf{x}^\alpha = \prod_{i=1}^d x_i^{\alpha_i}$ and $|\alpha| = \sum_{i=1}^d \alpha_i$. Learning the class $\mathcal{P}_d^M$ requires a networks with at least $\Omega\left(M^2\right)$ parameters (and this remains true even if we restrict to $O(1)$-bounded distributions). We show that for $O(1)$-bounded distributions, SGD learns $\mathcal{P}_c^M$, with error parameter $\epsilon$ (that is, it returns a predictor with error $\leq \epsilon$), using a network with $\tilde{O}\left(\frac{M^2}{\epsilon^2}\right)$ parameters and $O\left(\frac{M^2}{\epsilon^2}\right)$ SGD iterations.

**Learning Kernel Spaces**    Our result for polynomials is a corollary of a more general result about learning certain kernel spaces, that we describe next. Our result about memorization is not a direct corollary, but is also a refinement of that result. We consider the kernel $k : \mathbb{S}^{d-1} \times \mathbb{S}^{d-1} \to \mathbb{R}$ given by

$$k(\mathbf{x}, \mathbf{y}) = \langle \mathbf{x}, \mathbf{y} \rangle \cdot \mathop{\mathbb{E}}_{\mathbf{w} \sim \mathcal{N}(I, 0)} \sigma'\left(\langle \mathbf{w}, \mathbf{x} \rangle, \langle \mathbf{w}, \mathbf{y} \rangle\right) \tag{1}$$

which is a variant of the Neural Tangent Kernel Jacot et al. [2018]. We show that for $O(1)$-bounded distributions, SGD learns functions with norm $\leq M$ in the corresponding kernel space, with error parameter $\epsilon$, using a network with $\tilde{O}\left(\frac{M^2}{\epsilon^2}\right)$ parameters and $O\left(\frac{M^2}{\epsilon^2}\right)$ SGD iterations. We note that the network size is optimal up to the dependency on $\epsilon$ and poly-log factors, and the number of iteration is optimal up to a constant factor. This result is valid for most Lipschitz losses including the hinge loss and the log-loss, and for most popular activation functions, including the ReLU.

## 1.1   Related Work

The connection between networks, kernels and random features has a long history. Early work includes Williams [1997], Rahimi and Recht [2009]. In recent years, this connection was utilized to analyze neural networks algorithm (e.g. Andoni et al. [2014], Daniely et al. [2016], Daniely [2017], Cao and Gu [2019], Ji and Telgarsky [2019], Zou and Gu [2019], Ma et al. [2019], Du et al. [2018a], Arora et al. [2019], Song and Yang [2019], Oymak and Soltanolkotabi [2019a], Ge et al. [2019]). In fact, the vast majority of known non-linear learnable models, including memorization models, polynomials, and kernel spaces utilize this connection. It is worth mentioning very recent papers Daniely and Malach [2020], Yehudai and Shamir [2019], Allen-Zhu and Li [2019], Ghorbani et al. [2019] that proves learnability beyond NTK.

It is hard to quantitatively compare the various result about learning polynomials and kernels, as they often depend on various parameters of the distributions, and talk about different kernels and polynomial spaces. Yet, to the best of our knowledge, in none of the known results the network size is optimal in both the input dimension and the kernel norm as theorems 5 and 6. In this regard, we would like to mention Ji and Telgarsky [2019] which has optimal (logarithmic) dependence on $\epsilon$ in the case that the input distribution is realizable with margin in the NTK space. This should be

compared to our dependance which is on one hand quadratic in $1/\epsilon$, but on the other hand valid for both the realizable and un-realizable settings.

As for memorization results, as mentioned above, results with about near optimal network size either consider linearly separable data Brutzkus et al. [2018] or quadratic activation Ge et al. [2019]. As for non-polynomial activations and non-linearly-separable data, the results of Daniely [2017] imply that under rather mild conditions, $m$ points with arbitrary labels can be memorized by networks of size $\text{poly}(m)$, but without an exact specification of the exponent of the polynomial. Allen-Zhu et al. [2018] showed a memorization result using $\tilde{O}(m^{24})$ parameters. Zou and Gu [2019] improved the bound to $\tilde{O}(m^8)$, then to $\tilde{O}(m^6)$ by Du et al. [2018a] and Wu et al. [2019], to $\tilde{O}(m^4)$ by Du et al. [2018b], and finally, the state of the art until our work was memorization with $\tilde{O}(m^2)$ parameters Song and Yang [2019], Oymak and Soltanolkotabi [2019b]. We would also like to mention Fiat et al. [2019] whose result shares some ideas with our proof. In their paper it is shown that for the ReLU activation, linear optimization over the embedding $\Psi_{W,\mathbf{u}}$ can memorize $m$ points with $\tilde{O}(m)$ parameters.

## 2 Preliminaries

### 2.1 Notation

We denote vectors by bold-face letters (e.g. $\mathbf{x}$), matrices by upper case letters (e.g. $W$), and collection of matrices by bold-face upper case letters (e.g. $\mathbf{W}$). We denote the $i$'s row in a matrix $W$ by $\mathbf{w}_i$. The $p$-norm of $\mathbf{x} \in \mathbb{R}^d$ is denoted by $\|\mathbf{x}\|_p = \left( \sum_{i=1}^d |x_i|^p \right)^{\frac{1}{p}}$, and for a matrix $W$, $\|W\|$ is the spectral norm $\|W\| = \max_{\|\mathbf{x}\|=1} \|W\mathbf{x}\|$. We will also use the convention that $\|\mathbf{x}\| = \|\mathbf{x}\|_2$. For a distribution $\mathcal{D}$ on a space $\mathcal{X}$, $p \geq 1$ and $f : \mathcal{X} \to \mathbb{R}$ we denote $\|f\|_{p,\mathcal{D}} = (\mathbb{E}_{x \sim \mathcal{D}} |f(x)|^p)^{\frac{1}{p}}$. We denote by $L^2(\mathcal{X}, \mathbb{R}^d)$ the space of functions $f : \mathcal{X} \to \mathbb{R}^d$ with $\mathbb{E}_{x \sim \mathcal{D}} \|f(x)\|^2 < \infty$. Note that it is an inner product space w.r.t. the inner product $\langle f, g \rangle_{L^2(\mathcal{X}, \mathbb{R}^d)} = \mathbb{E}_{x \sim \mathcal{D}} \langle f(x), g(x) \rangle$. We use $\tilde{O}$ to hide poly-log factors.

### 2.2 Supervised learning

The goal in supervised learning is to devise a mapping from the input space $\mathcal{X}$ to an output space $\mathcal{Y}$ based on a sample $S = \{(\mathbf{x}_1, y_1), \ldots, (\mathbf{x}_m, y_m)\}$, where $(\mathbf{x}_i, y_i) \in \mathcal{X} \times \mathcal{Y}$ drawn i.i.d. from a distribution $\mathcal{D}$ over $\mathcal{X} \times \mathcal{Y}$. In our case, the instance space will always be $\mathbb{S}^{d-1}$. A supervised learning problem is further specified by a loss function $\ell : \mathbb{R} \times \mathcal{Y} \to [0, \infty)$, and the goal is to find a predictor $h : \mathcal{X} \to \mathbb{R}$ whose loss, $\mathcal{L}_{\mathcal{D}}(h) := \mathbb{E}_{(\mathbf{x},y) \sim \mathcal{D}} \ell(h(\mathbf{x}), y)$, is small. The *empirical* loss $\mathcal{L}_S(h) := \frac{1}{m} \sum_{i=1}^m \ell(h(\mathbf{x}_i), y_i)$ is commonly used as a proxy for the loss $\mathcal{L}_{\mathcal{D}}$. When $h$ is defined by a vector $\mathbf{w}$ of parameters, we will use the notations $\mathcal{L}_{\mathcal{D}}(\mathbf{w}) = \mathcal{L}_{\mathcal{D}}(h)$, $\mathcal{L}_S(\mathbf{w}) = \mathcal{L}_S(h)$ and $\ell_{(\mathbf{x},y)}(\mathbf{w}) = \ell(h(\mathbf{x}), y)$. For a class $\mathcal{H}$ of predictors from $\mathcal{X}$ to $\mathbb{R}$ we denote $\mathcal{L}_{\mathcal{D}}(\mathcal{H}) = \inf_{h \in \mathcal{H}} \mathcal{L}_{\mathcal{D}}(h)$ and $\mathcal{L}_S(\mathcal{H}) = \inf_{h \in \mathcal{H}} \mathcal{L}_S(h)$

A loss $\ell$ is $L$-Lipschitz if for all $y \in \mathcal{Y}$, the function $\ell_y(\hat{y}) := \ell(\hat{y}, y)$ is $L$-Lipschitz. Likewise, it is convex if $\ell_y$ is convex for every $y \in \mathcal{Y}$. We say that $\ell$ is $L$-*decent* if for every $y \in \mathcal{Y}$, $\ell_y$ is convex, $L$-Lipschitz, and twice differentiable in all but finitely many points.

### 2.3 Neural network learning

We will consider fully connected neural networks of depth 2 with $2q$ hidden neurons and activation function $\sigma : \mathbb{R} \to \mathbb{R}$. Throughout, we assume that the activation function is continuous, is twice differentiable in all but finitely many points, and there is $M > 0$ such that $|\sigma'(x)|, |\sigma''(x)| \leq M$ for every point $x \in \mathbb{R}$ for which $f$ is twice differentiable in $x$. We call such an activation a *decent* activation. This includes most popular activations, including the ReLU activation $\sigma(x) = \max(0, x)$, as well as most sigmoids.

Denote $\mathcal{N}_{d,q}^\sigma = \left\{ h_{\mathbf{W}}(\mathbf{x}) = \langle \mathbf{u}, \sigma(W\mathbf{x}) \rangle : W \in M_{2q,d}, \mathbf{u} \in \mathbb{R}^{2q} \right\}$. We also denote by $\mathbf{W} = (W, \mathbf{u})$ the aggregation of all weights. We next describe the learning algorithm that we analyze in this paper. We will use a variant of the popular Xavier initialization [Glorot and Bengio, 2010] for the network

weights, which we call *Xavier initialization with zero outputs*. The neurons will be arranged in pairs, where each pair consists of two neurons that are initialized identically, up to sign. Concretely, the weight matrix $W$ will be initialized to be a duplication $W = \begin{bmatrix} W' \\ W' \end{bmatrix}$ of a matrix $W'$ of standard Gaussians[2] and $\mathbf{u}$ will be a duplication of the all-$B$ vector in dimension $q$, for some $B > 0$, with its negation. We denote the distribution of this initialization scheme by $\mathcal{I}(d, q, B)$. Note that if $\mathbf{W} \sim \mathcal{I}(d, q, B)$ then w.p. 1, $\forall \mathbf{x}$, $h_{\mathbf{W}}(\mathbf{x}) = 0$. Finally, the training algorithm is described in 1.

---

**Algorithm 1** Neural Network Training

---

**Input:** Network parameters $\sigma$ and $d, q$, loss $\ell$, initialization parameter $B > 0$, learning rate $\eta > 0$, batch size $b$, number of steps $T > 0$, access to samples from a distribution $\mathcal{D}$
Sample $\mathbf{W}^1 \sim \mathcal{I}(d, q, B)$
**for** $t = 1, \ldots, T$ **do**
    Obtain a mini-batch $S_t = \{(\mathbf{x}_i^t, y_i^t)\}_{i=1}^b \sim \mathcal{D}^b$
    With back-propagation, calculate a stochastic gradient $\nabla \mathcal{L}_{S_t}(\mathbf{W}^t)$ and update $\mathbf{W}^{t+1} = \mathbf{W}^t - \eta \nabla \mathcal{L}_{S_t}(\mathbf{W}^t)$
**end for**
Choose $t \in [T]$ uniformly at random and return $\mathbf{W}_t$

---

## 2.4 Kernel spaces

Let $\mathcal{X}$ be a set. A *kernel* is a function $k : \mathcal{X} \times \mathcal{X} \to \mathbb{R}$ such that for every $x_1, \ldots, x_m \in \mathcal{X}$ the matrix $\{k(x_i, x_j)\}_{i,j}$ is positive semi-definite. A *kernel space* is a Hilbert space $\mathcal{H}$ of functions from $\mathcal{X}$ to $\mathbb{R}$ such that for every $x \in \mathcal{X}$ the linear functional $f \in \mathcal{H} \mapsto f(x)$ is bounded. The following theorem describes a one-to-one correspondence between kernels and kernel spaces.

**Theorem 1.** *For every kernel $k$ there exists a unique kernel space $\mathcal{H}_k$ such that for every $x, x' \in \mathcal{X}$, $k(x, x') = \langle k(\cdot, x), k(\cdot, x') \rangle_{\mathcal{H}_k}$. Likewise, for every kernel space $\mathcal{H}$ there is a kernel $k$ for which $\mathcal{H} = \mathcal{H}_k$.*

We denote the norm and inner product in $\mathcal{H}_k$ by $\| \cdot \|_k$ and $\langle \cdot, \cdot \rangle_k$. The following theorem describes a tight connection between kernels and embeddings of $X$ into Hilbert spaces.

**Theorem 2.** *A function $k : \mathcal{X} \times \mathcal{X} \to \mathbb{R}$ is a kernel if and only if there exists a mapping $\Psi : \mathcal{X} \to \mathcal{H}$ to some Hilbert space for which $k(x, x') = \langle \Psi(x), \Psi(x') \rangle_{\mathcal{H}}$. In this case, $\mathcal{H}_k = \{f_{\Psi, \mathbf{v}} \mid \mathbf{v} \in \mathcal{H}\}$ where $f_{\Psi, \mathbf{v}}(x) = \langle \mathbf{v}, \Psi(x) \rangle_{\mathcal{H}}$. Furthermore, $\|f\|_k = \min\{\|\mathbf{v}\|_{\mathcal{H}} : f_{\Psi, \mathbf{v}}\}$ and the minimizer is unique.*

For a kernel $k$ and $M > 0$ we denote $\mathcal{H}_k^M = \{h \in \mathcal{H}_k : \|h\|_k \leq M\}$. We note that spaces of the form $\mathcal{H}_k^M$ often form a benchmark for learning algorithms.

## 2.5 The Neural Tangent Kernel

Fix network parameters $\sigma, d, q$ and $B$. The *neural tangent kernel* corresponding to weights $\mathbf{W}$ is[3]

$$\mathrm{tk}_{\mathbf{W}}(\mathbf{x}, \mathbf{y}) = \frac{\langle \nabla_{\mathbf{W}} h_{\mathbf{W}}(\mathbf{x}), \nabla_{\mathbf{W}} h_{\mathbf{W}}(\mathbf{y}) \rangle}{2qB^2}$$

The neural tangent kernel space, $\mathcal{H}_{\mathrm{tk}_{\mathbf{W}}}$, is a linear approximation of the trajectories in which $h_W$ changes by changing $W$ a bit. Specifically, $h \in \mathcal{H}_{\mathrm{tk}_{\mathbf{W}}}$ if and only if there is $\mathbf{U}$ such that

$$\forall \mathbf{x} \in \mathbb{S}^{d_1 - 1}, \quad h(\mathbf{x}) = \lim_{\epsilon \to 0} \frac{h_{\mathbf{W} + \epsilon \mathbf{U}}(\mathbf{x}) - h_{\mathbf{W}}(\mathbf{x})}{\epsilon} \tag{2}$$

Furthermore, we have that $\sqrt{q}B \cdot \|h\|_{\mathrm{tk}\mathbf{W}}$ is the minimal Euclidean norm of $\mathbf{U}$ that satisfies equation (2). The *expected initial neural tangent kernel* is

$$\mathrm{tk}_{\sigma,B}(\mathbf{x},\mathbf{y}) = \mathrm{tk}_{\sigma,d,q,B}(\mathbf{x},\mathbf{y}) = \mathop{\mathbb{E}}_{\mathbf{W}\sim(d,q,B)} \mathrm{tk}_{\mathbf{W}}(\mathbf{x},\mathbf{y})$$

We will later see that $\mathrm{tk}_{\sigma,d,q,B}$ depends only on $\sigma$ and $B$. If the network is large enough, we can expect that at the onset of the optimization process, $\mathrm{tk}_{\sigma,B} \approx k_{\mathbf{W}}$. Hence, approximately, $\mathcal{H}_{\mathrm{tk}_{\sigma,B}}$ consists of the directions in which the initial function computed by the network can move. Since the initial function (according to Xavier initialization with zero outputs) is 0, $\mathcal{H}_{\mathrm{tk}_{\sigma,B}}$ is a linear approximation of the space of functions computed by the network in the vicinity of the initial weights. NTK theory based of the fact close enough to the initialization point, the linear approximation is good, and hence SGD on NN can learn functions in $\mathcal{H}_{\mathrm{tk}_{\sigma,B}}$ that has sufficiently small norm. The main question is how small should the norm be, or alternatively, how large should the network be.

We next derive a formula for $\mathrm{tk}_{\sigma,B}$. We have, for $\mathbf{W} \sim \mathcal{I}(d,q,B)$

$$
\begin{aligned}
\mathrm{tk}_{\mathbf{W}}(\mathbf{x},\mathbf{y}) &= \frac{\langle \nabla_{\mathbf{W}} h_{\mathbf{W}}(\mathbf{x}), \nabla_{\mathbf{W}} h_{\mathbf{W}}(\mathbf{y}) \rangle}{2qB^2} \\
&= \frac{1}{qB^2}\sum_{i=1}^{q}\langle B\sigma'(\langle \mathbf{w}_i,\mathbf{x}\rangle)\mathbf{x}, B\sigma'(\langle \mathbf{w}_i,\mathbf{y}\rangle)\mathbf{y}\rangle + \frac{1}{qB^2}\sum_{i=1}^{q}\sigma(\langle \mathbf{w}_i,\mathbf{x}\rangle)\sigma(\langle \mathbf{w}_i,\mathbf{y}\rangle) \\
&= \frac{\langle \mathbf{x},\mathbf{y}\rangle}{q}\sum_{i=1}^{q}\sigma'(\langle \mathbf{w}_i,\mathbf{x}\rangle)\sigma'(\langle \mathbf{w}_i,\mathbf{y}\rangle) + \frac{1}{qB^2}\sum_{i=1}^{q}\sigma(\langle \mathbf{w}_i,\mathbf{x}\rangle)\sigma(\langle \mathbf{w}_i,\mathbf{y}\rangle)
\end{aligned}
$$

Taking expectation we get

$$\mathrm{tk}_{\sigma,B}(\mathbf{x},\mathbf{y}) = \langle \mathbf{x},\mathbf{y}\rangle\,\hat{\sigma}'(\langle \mathbf{x},\mathbf{y}\rangle) + \frac{1}{B^2}\hat{\sigma}(\langle \mathbf{x},\mathbf{y}\rangle) = \langle \mathbf{x},\mathbf{y}\rangle\,k_{\sigma'}(\mathbf{x},\mathbf{y}) + \frac{1}{B^2}k_{\sigma}(\mathbf{x},\mathbf{y})$$

Finally, we decompose the expected initial neural tangent kernel into two kernels, that corresponds to the hidden and output weights respectively. Namely, we let

$$\mathrm{tk}_{\sigma,B} = \mathrm{tk}_{\sigma,B}^{h} + \mathrm{tk}_{\sigma,B}^{o} \text{ for } \mathrm{tk}_{\sigma}^{h}(\mathbf{x},\mathbf{y}) = \langle \mathbf{x},\mathbf{y}\rangle\,\hat{\sigma}'(\langle \mathbf{x},\mathbf{y}\rangle) \text{ and } \mathrm{tk}_{\sigma,B}^{o}(\mathbf{x},\mathbf{y}) = \frac{1}{B^2}\hat{\sigma}(\langle \mathbf{x},\mathbf{y}\rangle)$$

Accordingly, we denote

$$\mathrm{tk}_{\mathbf{W}}^{h}(\mathbf{x},\mathbf{y}) = \frac{\langle \mathbf{x},\mathbf{y}\rangle}{q}\sum_{i=1}^{q}\sigma'(\langle \mathbf{w}_i,\mathbf{x}\rangle)\sigma'(\langle \mathbf{w}_i,\mathbf{y}\rangle) \text{ and } \mathrm{tk}_{\mathbf{W}}^{o}(\mathbf{x},\mathbf{y}) = \frac{1}{qB^2}\sum_{i=1}^{q}\sigma(\langle \mathbf{w}_i,\mathbf{x}\rangle)\sigma(\langle \mathbf{w}_i,\mathbf{y}\rangle)$$

### 2.6 Vector Random Feature Schemes

Random features schemes Williams [1997], Rahimi and Recht [2009] introduced as a mean for developing fast algorithm for learning kernel spaces. Here, we will use random features as a tool for analyzing SGD on networks. Let $\mathcal{X}$ be a measurable space and let $k : \mathcal{X} \times \mathcal{X} \to \mathbb{R}$ be a kernel. A *random features scheme* (RFS) for $k$ is a pair $(\psi,\mu)$ where $\mu$ is a probability measure on a measurable space $\Omega$, and $\psi : \Omega \times \mathcal{X} \to \mathbb{R}^d$ is a measurable function such that

$$\forall \mathbf{x},\mathbf{x}' \in \mathcal{X}, \quad k(\mathbf{x},\mathbf{x}') = \mathop{\mathbb{E}}_{\omega\sim\mu}\left[\langle \psi(\omega,\mathbf{x}), \psi(\omega,\mathbf{x}')\rangle\right]. \tag{3}$$

We often refer to $\psi$ (rather than $(\psi,\mu)$) as the RFS. Our motivation form considering vector RFS in this paper steams from the *NTK RFS*, which is given by the mapping $\psi : \mathbb{R}^d \times \mathbb{S}^{d-1} \to \mathbb{R}^d$ defined by $\psi(\omega,\mathbf{x}) = \sigma'(\langle \omega,\mathbf{x}\rangle)\mathbf{x}$ and $\mu$ being the standard Gaussian measure on $\mathbb{R}^d$. Note that it is an RFS for the kernel $\mathrm{tk}_{\sigma}^{h}$ (see section 2.5).

We define the *norm* of $\psi$ as $\|\psi\| = \sup_{\omega,\mathbf{x}}\|\psi(\omega,\mathbf{x})\|$. We say that $\psi$ is *C-bounded* if $\|\psi\| \le C$. We say that an RFS $\psi : \Omega \times \mathbb{S}^{d-1} \to \mathbb{R}^d$ is *factorized* if there is a function $\psi' : \Omega \times \mathbb{S}^{d-1} \to \mathbb{R}$ such that $\psi(\omega,\mathbf{x}) = \psi'(\omega,\mathbf{x})\mathbf{x}$. We note that the NTK RFS is factorized and $C$-bounded for $C = \|\sigma'\|_{\infty}$.

Fix a $C$-bounded RFS $\psi$ for a kernel $k$. A *random q-embedding* generated from $\psi$ is the random mapping $\Psi_{\boldsymbol{\omega}}(\mathbf{x}) := \frac{(\psi(\omega_1,\mathbf{x}),\ldots,\psi(\omega_q,\mathbf{x}))}{\sqrt{q}}$, where $\omega_1,\ldots,\omega_q \sim \mu$ are i.i.d. The random *q-kernel*

corresponding to $\Psi_{\boldsymbol{\omega}}$ is $k_{\boldsymbol{\omega}}(\mathbf{x}, \mathbf{x}') = \langle \Psi_{\boldsymbol{\omega}}(\mathbf{x}), \Psi_{\boldsymbol{\omega}}(\mathbf{x}') \rangle$. Likewise, the random $q$-*kernel space* corresponding to $\Psi_{\boldsymbol{\omega}}$ is $\mathcal{H}_{k_{\boldsymbol{\omega}}}$. We note that in the case of the NTK RFS, a random $q$-embedding is, up to scaling, the gradient of a randomly initialized network. Likewise, $\mathrm{tk}_W^h$ is a random $q$-kernel generated from the NTK RFS.

It would be useful to consider the embedding

$$\mathbf{x} \mapsto \Psi^{\mathbf{x}} \quad \text{where} \quad \Psi^{\mathbf{x}} := \psi(\cdot, \mathbf{x}) \in L^2(\Omega, \mathbb{R}^d). \tag{4}$$

From (3) it holds that for any $\mathbf{x}, \mathbf{x}' \in \mathcal{X}$, $k(\mathbf{x}, \mathbf{x}') = \left\langle \Psi^{\mathbf{x}}, \Psi^{\mathbf{x}'} \right\rangle_{L^2(\Omega)}$. In particular, from Theorem 2, for every $f \in \mathcal{H}_k$ there is a unique function $\check{f} \in L^2(\Omega, \mathbb{R}^d)$ such that

$$\|\check{f}\|_{L^2(\Omega)} = \|f\|_k \tag{5}$$

and for every $\mathbf{x} \in \mathcal{X}$,

$$f(\mathbf{x}) = \left\langle \check{f}, \Psi^{\mathbf{x}} \right\rangle_{L^2(\Omega, \mathbb{R}^d)} = \mathop{\mathbb{E}}_{\omega \sim \mu} \left\langle \check{f}(\omega), \psi(\omega, \mathbf{x}) \right\rangle. \tag{6}$$

Let us denote $f_{\boldsymbol{\omega}}(\mathbf{x}) = \frac{1}{q} \sum_{i=1}^q \left\langle \check{f}(\omega_i), \psi(\omega_i, \mathbf{x}) \right\rangle$. From (6) we have that $\mathbb{E}_{\boldsymbol{\omega}}[f_{\boldsymbol{\omega}}(\mathbf{x})] = f(\mathbf{x})$. Furthermore, for every $\mathbf{x}$, the variance of $f_{\boldsymbol{\omega}}(\mathbf{x})$ is at most

$$\frac{1}{q} \mathop{\mathbb{E}}_{\omega \sim \mu} \left| \left\langle \check{f}(\omega), \psi(\omega, \mathbf{x}) \right\rangle \right|^2 \leq \frac{C^2}{q} \mathop{\mathbb{E}}_{\omega \sim \mu} \left| \check{f}(\omega) \right|^2 = \frac{C^2 \|f\|_k^2}{q}.$$

An immediate consequence is the following corollary.

**Corollary 3** (Function Approximation)**.** *For all* $\mathbf{x} \in \mathcal{X}$, $\mathbb{E}_{\boldsymbol{\omega}} |f(\mathbf{x}) - f_{\boldsymbol{\omega}}(\mathbf{x})|^2 \leq \frac{C^2 \|f\|_k^2}{q}$.

Now, if $\mathcal{D}$ is a distribution on $\mathcal{X}$ we get that

$$\mathop{\mathbb{E}}_{\boldsymbol{\omega}} \|f - f_{\boldsymbol{\omega}}\|_{2, \mathcal{D}} \overset{\text{Jensen}}{\leq} \sqrt{\mathop{\mathbb{E}}_{\boldsymbol{\omega}} \|f - f_{\boldsymbol{\omega}}\|_{2, \mathcal{D}}^2} = \sqrt{\mathop{\mathbb{E}}_{\boldsymbol{\omega}} \mathop{\mathbb{E}}_{\mathbf{x} \sim \mathcal{D}} |f(\mathbf{x}) - f_{\boldsymbol{\omega}}(\mathbf{x})|^2} = \sqrt{\mathop{\mathbb{E}}_{\mathbf{x}} \mathop{\mathbb{E}}_{\boldsymbol{\omega}} |f(\mathbf{x}) - f_{\boldsymbol{\omega}}(\mathbf{x})|^2} \leq \frac{C \|f\|_k}{\sqrt{q}}$$

Using the above inequality, it is possible to show that (see theorem 10 below) SGD on top of a random $q$-embedding, using a convex and Lipschitz loss, is guaranteed to find a function $\hat{f}$ that satisfies $\mathbb{E} \mathcal{L}_{\mathcal{D}}(\hat{f}) \leq \mathcal{L}_{\mathcal{D}}(f^*) + O\left( \frac{\|f^*\|_k}{\sqrt{q}} \right)$ for any $f^* \in \mathcal{H}_k$.

## 3 Results

We next present our results in detail. Due to lack of space, all proofs are differed to the appendix.

### 3.1 Verctor RFS and NTK Convergence

Fix a $C$-bounded RFS $\psi : \Omega \times \mathcal{X} \to \mathbb{R}^d$ for a kernel $k$. Corollary 3 implies that $O\left( \frac{\|f\|_k^2}{\epsilon^2} \right)$ random features suffices to guarantee that for every $f \in \mathcal{H}_k$, in expectation, the empirical kernel space will contain an $\epsilon$ approximation of $f$. This bound does not depend on $d$, the dimension of a single random feature. We might expect that at least in some cases, $d$-dimensional random feature is as good as $d$ one-dimensional random features. The next result show that for factorized $RFS$ and $O(1)$-bounded distributions this is indeed the case and $O\left( \frac{\|f\|_k^2}{d\epsilon^2} \right)$ random features suffices to guarantee $\epsilon$-approximation.

**Theorem 4.** *Assume that* $\psi : \Omega \times \mathbb{S}^{d-1} \to \mathbb{R}^d$ *is factorized and* $\mathcal{D}$ *is $R$-bounded distribution. Then,*

$$\mathop{\mathbb{E}}_{\boldsymbol{\omega}} \|f - f_{\boldsymbol{\omega}}\|_{2, \mathcal{D}} \leq \sqrt{\mathop{\mathbb{E}}_{\boldsymbol{\omega}} \|f - f_{\boldsymbol{\omega}}\|_{2, \mathcal{D}}^2} \leq \frac{RC \|f\|_k}{\sqrt{qd}}$$

*Furthermore, if* $\ell : \mathbb{S}^{d-1} \times Y \to [0, \infty)$, *is $L$-Lipschitz loss and $\mathcal{D}'$ is a distribution of $\mathbb{S}^{d-1} \times Y$ with $R$-bounded marginal then* $\mathbb{E}_{\boldsymbol{\omega}} \mathcal{L}_{\mathcal{D}'}(f_{\boldsymbol{\omega}}) \leq \mathcal{L}_{\mathcal{D}'}(f) + \frac{LRC \|f\|_k}{\sqrt{qd}}$

Using the above inequality, it is possible to show that (see theorem 10 below) SGD on top of a random $q$-embedding, using a convex and Lipschitz loss, is guaranteed to find a function $\hat{f}$ that satisfies $\mathbb{E}\,\mathcal{L}_\mathcal{D}(\hat{f}) \le \mathcal{L}_\mathcal{D}(f^*) + O\left(\frac{\|f^*\|_k}{\sqrt{qd}}\right)$ for any $f^* \in \mathcal{H}_k$. Applying this to the NTK RFS, and via further reduction to neural network learning, we can show that a similar guarantee is valid for algorithm 1. This is described in the next section.

## 3.2 Learning the neural tangent kernel space with SGD on NN

Fix a decent activation function $\sigma$ and a decent loss $\ell$. We shows that algorithm 1 can learn the class $\mathcal{H}^M_{\mathrm{tk}^h_\sigma}$ using a network with $\tilde{O}\left(\frac{M^2}{\epsilon^2}\right)$ parameters and using $O\left(\frac{M^2}{\epsilon^2}\right)$ examples. We note that unless $\sigma$ is linear, the number of samples is optimal up to constant factor, and the number of parameters is optimal, up to poly-log factor and the dependency on $\epsilon$. This remains true even if we restrict to $O(1)$-bounded distributions.

**Theorem 5.** *Given $d$, $M > 0$, $R > 0$ and $\epsilon > 0$ there is a choice of $q = \tilde{O}\left(\frac{M^2 R^2}{d\epsilon^2}\right)$, $T = O\left(\frac{M^2}{\epsilon^2}\right)$, as well as $B > 0$ and $\eta > 0$, such that for every $R$-bounded distribution $\mathcal{D}$ and batch size $b$, the function $h$ returned by algorithm 1 satisfies $\mathbb{E}\,\mathcal{L}_\mathcal{D}(h) \le \mathcal{L}_\mathcal{D}\left(\mathcal{H}^M_{\mathrm{tk}^h_\sigma}\right) + \epsilon$*

As an application, we conclude that for the ReLU activation, algorithm 1 can learn even polynomials of bounded norm with near optimal sample complexity and network size. We denote

$$\mathcal{P}^M_c = \left\{ p(\mathbf{x}) = \sum_{|\alpha| \text{ is even and } \le c} a_\alpha \mathbf{x}^\alpha : \sum_{|\alpha| \text{ is even and } \le c} a_\alpha^2 \le M^2 \right\}$$

For the ReLU activation $\sigma$, it holds that for every constant $c$, $\mathcal{P}^M_c \subset \mathcal{H}^{O(M)}_{\mathrm{tk}^h_\sigma}$ (e.g. Daniely et al. [2016]). Theorem 5 therefore implies that

**Theorem 6.** *Fix a constant $c > 0$ and assume that the activation is ReLU. Given $d$, $M > 0$, $R > 0$ and $\epsilon > 0$ there is a choice of $q = \tilde{O}\left(\frac{M^2 R^2}{d\epsilon^2}\right)$, $T = O\left(\frac{M^2}{\epsilon^2}\right)$, as well as $B > 0$ and $\eta > 0$, such that for every $R$-bounded distribution $\mathcal{D}$ and batch size $b$, the function $h$ returned by algorithm 1 satisfies $\mathbb{E}\,\mathcal{L}_\mathcal{D}(h) \le \mathcal{L}_\mathcal{D}\left(\mathcal{P}^M_c\right) + \epsilon$*

We note that as in theorem 5, the number of samples is optimal up to constant factor, and the number of parameters is optimal, up to poly-log factor and the dependency on $\epsilon$, and this remains true even if we restrict to $O(1)$-bounded distributions.

## 3.3 Memorization

Theorem 5 can be applied to analyze memorization by SGD. Assume that $\ell$ is the hinge loss (similar result is valid for many other losses such as the log-loss) and $\sigma$ is any decent non-linear activation. Let $S = \{(\mathbf{x}_1, y_1), \ldots, (\mathbf{x}_m, y_m)\}$ be $m$ random, independent and uniform points in $\mathbb{S}^{d-1} \times \{\pm 1\}$ with $m = d^c$ for some $c > 1$. Suppose that we run SGD on top of $S$. Namely, we run algorithm 1 where the underlying distribution is the uniform distribution on the points in $S$. Let $h : \mathbb{S}^{d-1} \to \mathbb{R}$ be the output of the algorithm. We say that the algorithm memorized the $i$'th example if $y_i h(\mathbf{x}_i) > 0$. The memorization problem investigate how many points the algorithm can memorize, were most of the focus is on how large the network should be in order to memorize $1 - \epsilon$ fraction of the points.

As shown in section A.5, the uniform distribution on the examples in $S$ is $(1 + o(1))$-bounded w.h.p. over the choice of $S$. Likewise, it is not hard to show that w.h.p. over the choice of $S$ there is a function $h^* \in \mathcal{H}^{O(m)}_k$ such that $h^*(\mathbf{x}_i) = y_i$ for all $i$. By theorem 5 we can conclude the by running SGD on a network with $\tilde{O}\left(\frac{m}{\epsilon^2}\right)$ parameters and $O\left(\frac{m}{\epsilon^2}\right)$ steps, the network will memorize $1 - \epsilon$ fraction of the points. This size of networks is optimal up to poly-log factors, and the dependency of $\epsilon$. This is satisfactory is $\epsilon$ is considered a constant. However, for small $\epsilon$, more can be desired. For instance, in the case that we want to memorize all points, we need $\epsilon < \frac{1}{m}$, and we get a network with $m^3$ parameters. To circumvent that, we perform a more refined analysis of this memorization problem and show that even perfect memorization of $m$ points can be done via SGD on a network with $\tilde{O}(m)$ parameters, which is optimal, up to poly-log factors.

**Theorem 7.** *There is a choice of $q = \tilde{O}\left(\frac{m}{d}\right)$, $T = \tilde{O}\left(\frac{m}{\epsilon^2}\right)$, as well as $B > 0$ and $\eta > 0$, such that for every batch size $b$, w.p. $1 - o_m(1)$, the function $h$ returned by algorithm 1 memorizes $1 - \epsilon$ fraction of the examples.*

We emphasize the our result is true for any non-linear and decent activation function.

### 3.4 Open Questions

The most obvious open question is to generalize our results to the standard Xavier initialization, where $W$ is a matrix of independent standeard Gaussians, while $\mathbf{u}$ is a vector of independent centered Gaussians of variance $\frac{1}{q}$. Another open question is to generalize our result to deeper networks.

## Broader Impact

Not applicable as far as we can see (this is a purely theoretical paper).

## Acknowledgments and Disclosure of Funding

This research is partially supported by ISF grant 2258/19

## Footnotes

[1]More specifically, we mean that the proofs require number of parameters that is suboptimal by a multiplicative factor that grows polynomially with one of the problem parameters – either the model capacity (margin, VC dimension, etc.), the desired error (i.e. $\epsilon$), or the input dimension.

[2]It is more standard to assume that the instances has $L^2$ norm $O\left(\sqrt{d}\right)$, or infinity norm $O(1)$, and the entries of $W'$ has variance $\frac{1}{d}$. For the sake of notational convenience we chose a different scaling—divided the instances by $\sqrt{d}$ and accordingly multiplied the initial matrix by $\sqrt{d}$. Identical results can be derived for the more standard convention.

[3]The division by $2qB^2$ is for notational convenience.

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
