[Supplementary Material]

# Memorizing Gaussians with no over-parameterizaion via gradient decent on neural networks

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

}'\left(\langle \mathbf{x},\mathbf{y}\rangle\right) + \frac{1}{B^2}\hat{\sigma}\left(\langle \mathbf{x},\mathbf{y}\rangle\right) = \langle \mathbf{x},\mathbf{y}\rangle\, k_{\sigma'}(\mathbf{x},\mathbf{y}) + \frac{1}{B^2}k_{\sigma}(\mathbf{x},\mathbf{y})
$$

194   Finally, we decompose the expected initial neural tangent kernel into two kernels, that corresponds to
195   the hidden and output weights respectively. Namely, we let

$$
\mathrm{tk}_{\sigma,B} = \mathrm{tk}_{\sigma,B}^{h} + \mathrm{tk}_{\sigma,B}^{o} \text{ for } \mathrm{tk}_{\sigma}^{h}(\mathbf{x},\mathbf{y}) = \langle \mathbf{x},\mathbf{y}\rangle\,\hat{\sigma}'\left(\langle \mathbf{x},\mathbf{y}\rangle\right) \text{ and } \mathrm{tk}_{\sigma,B}^{o}(\mathbf{x},\mathbf{y}) = \frac{1}{B^2}\hat{\sigma}\left(\langle \mathbf{x},\mathbf{y}\rangle\right)
$$

196   Accordingly, we denote

$$
\mathrm{tk}_{\mathbf{W}}^{h}(\mathbf{x},\mathbf{y}) = \frac{\langle \mathbf{x},\mathbf{y}\rangle}{q}\sum_{i=1}^{q}\sigma'\left(\langle \mathbf{w}_i,\mathbf{x}\rangle\right)\sigma'\left(\langle \mathbf{w}_i,\mathbf{y}\rangle\right) \text{ and } \mathrm{tk}_{\mathbf{W}}^{o}(\mathbf{x},\mathbf{y}) = \frac{1}{qB^2}\sum_{i=1}^{q}\sigma\left(\langle \mathbf{w}_i,\mathbf{x}\rangle\right)\sigma\left(\langle \mathbf{w}_i,\mathbf{y}\rangle\right)
$$

### 2.6   Vector Random Feature Schemes

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

## A  Proofs

### A.1  More preliminaries: inner product kernels and Hermite polynomials

A special type of kernels that we will useful for us are *inner product kernels*. These are kernels $k : \mathbb{S}^{d-1} \times \mathbb{S}^{d-1} \to \mathbb{R}$ of the form

$$k(\mathbf{x}, \mathbf{y}) = \sum_{n=0}^{\infty} b_n \langle \mathbf{x}, \mathbf{y} \rangle^n$$

For scalars $b_n \geq 0$ with $\sum_{n=0}^{\infty} b_n < \infty$. It is well known that for any such sequence $k$ is a kernel. The following lemma summarizes a few properties of inner product kernels.

**Lemma 8.** *Let $k$ be the inner product kernel $k(\mathbf{x}, \mathbf{y}) = \sum_{n=0}^{\infty} b_n \langle \mathbf{x}, \mathbf{y} \rangle^n$. Suppose that $b_n > 0$*

    *1. If $p(\mathbf{x}) = \sum_{|\alpha|=n} a_\alpha \mathbf{x}^\alpha$ then $p \in \mathcal{H}_k$ and furthermore $\|p\|_k^2 \leq \frac{1}{b_n} \sum_{|\alpha|=n} a_\alpha^2$*

    *2. For every $\mathbf{u} \in \mathbb{S}^{d-1}$, the function $f(\mathbf{x}) = \langle \mathbf{u}, \mathbf{x} \rangle^n$ belongs to $\mathcal{H}_k$ and $\|f\|_k^2 = \frac{1}{b_n}$*

Hermite polynomials $h_0, h_1, h_2, \dots$ are the sequence of orthonormal polynomials corresponding to the standard Gaussian measure on $\mathbb{R}$. Fix an activation $\sigma : \mathbb{R} \to \mathbb{R}$. Following the terminology of [9] we define the *dual activation* of $\sigma$ as

$$\hat{\sigma}(\rho) = \mathop{\mathbb{E}}_{X,Y \text{ are } \rho\text{-correlated standard Gaussian}} \sigma(X)\sigma(Y)$$

It holds that if $\sigma = \sum_{n=0}^{\infty} a_n h_n$ then

$$\hat{\sigma}(\rho) = \sum_{n=0}^{\infty} a_n^2 \rho^n$$

In particular, $k_\sigma(\mathbf{x}, \mathbf{y}) := \hat{\sigma}(\langle \mathbf{x}, \mathbf{y} \rangle)$ is an inner product kernel.

### A.2  Vector random feature schemes

For the rest of this section, let us fix a $C$-bounded RFS $\psi$ for a kernel $k$ and a random $q$ embedding $\Psi_{\boldsymbol{\omega}}$. For every $\mathbf{x}, \mathbf{x}' \in \mathcal{X}$

$$k_{\boldsymbol{\omega}}(\mathbf{x}, \mathbf{x}') = \frac{1}{q} \sum_{i=1}^{q} \langle \psi(\omega_i, \mathbf{x}), \psi(\omega_i, \mathbf{x}') \rangle$$

is an average of $q$ independent random variables whose expectation is $k(\mathbf{x}, \mathbf{x}')$. By Hoeffding's bound we have.

**Theorem 9** (Kernel Approximation). *Assume that $q \geq \frac{2C^4 \log\left(\frac{2}{\delta}\right)}{\epsilon^2}$, then for every $\mathbf{x}, \mathbf{x}' \in \mathcal{X}$ we have $\Pr\left(|k_{\boldsymbol{\omega}}(\mathbf{x}, \mathbf{x}') - k(\mathbf{x}, \mathbf{x}')| \geq \epsilon\right) \leq \delta$.*

We next discuss approximation of functions in $\mathcal{H}_k$ by functions in $\mathcal{H}_{k_{\boldsymbol{\omega}}}$, and prove theorem 3

379 *Proof.* (of theorem 4) Let $\mathbf{x} \sim \mathcal{D}$ and $\omega \sim \mu$. We have

$$
\underset{\boldsymbol{\omega}}{\mathbb{E}} \, \|f - f_{\boldsymbol{\omega}}\|_{2,\mathcal{D}} \overset{\text{Jensen's Inequality}}{\leq} \sqrt{\underset{\boldsymbol{\omega}}{\mathbb{E}} \, \|f - f_{\boldsymbol{\omega}}\|_{2,\mathcal{D}}^2}
$$

$$
= \sqrt{\underset{\boldsymbol{\omega}}{\mathbb{E}} \underset{\mathbf{x}}{\mathbb{E}} \, |f(\mathbf{x}) - f_{\boldsymbol{\omega}}(\mathbf{x})|^2}
$$

$$
= \sqrt{\underset{\mathbf{x}}{\mathbb{E}} \underset{\boldsymbol{\omega}}{\mathbb{E}} \, |f(\mathbf{x}) - f_{\boldsymbol{\omega}}(\mathbf{x})|^2}
$$

$$
= \sqrt{\frac{\mathbb{E}_{\mathbf{x}} \, \mathbb{E}_{\omega \sim \mu} \, \left| \langle \check{f}(\omega), \psi(\omega, \mathbf{x}) \rangle - f(\mathbf{x}) \right|^2}{q}}
$$

$$
\overset{\text{Variance is bounded by squared } L^2 \text{ norm}}{\leq} \sqrt{\frac{\mathbb{E}_{\mathbf{x}} \, \mathbb{E}_{\omega \sim \mu} \, \left| \langle \check{f}(\omega), \psi(\omega, \mathbf{x}) \rangle \right|^2}{q}}
$$

$$
= \sqrt{\frac{\mathbb{E}_{\omega \sim \mu} \, \mathbb{E}_{\mathbf{x}} \, \left| \langle \check{f}(\omega), \psi'(\omega, \mathbf{x})\mathbf{x} \rangle \right|^2}{q}}
$$

$$
\overset{\psi \text{ and hence also } \psi' \text{ is } C\text{-bounded}}{\leq} C \sqrt{\frac{\mathbb{E}_{\omega \sim \mu} \, \mathbb{E}_{\mathbf{x}} \, \left| \langle \check{f}(\omega), \mathbf{x} \rangle \right|^2}{q}}
$$

$$
\overset{\mathcal{D} \text{ is } R\text{-bounded}}{\leq} CR \sqrt{\frac{\mathbb{E}_{\omega \sim \mu} \, \left\| \check{f}(\omega) \right\|^2}{qd}}
$$

$$
\overset{\text{Equation (5)}}{=} \frac{CR\|f\|_k}{\sqrt{qd}} \, .
$$

380 Finally, for $L$-Lipschitz $\ell$, and $(\mathbf{x}, y) \sim \mathcal{D}'$ then

$$
\underset{\boldsymbol{\omega}}{\mathbb{E}} \, L_{\mathcal{D}'}(f_{\boldsymbol{\omega}}) = \underset{\boldsymbol{\omega}}{\mathbb{E}} \underset{\mathbf{x},y}{\mathbb{E}} \, \ell(f_{\boldsymbol{\omega}}(\mathbf{x}), y)
$$

$$
\leq \underset{\boldsymbol{\omega}}{\mathbb{E}} \underset{\mathbf{x},y}{\mathbb{E}} \, \ell(f(\mathbf{x}), y) + L \underset{\boldsymbol{\omega}}{\mathbb{E}} \underset{\mathbf{x}}{\mathbb{E}} \, |f(\mathbf{x}) - f_{\boldsymbol{\omega}}(\mathbf{x})|
$$

$$
= \underset{\mathbf{x},y}{\mathbb{E}} \, \ell(f(\mathbf{x}), y) + L \underset{\boldsymbol{\omega}}{\mathbb{E}} \underset{\mathbf{x}}{\mathbb{E}} \, |f(\mathbf{x}) - f_{\boldsymbol{\omega}}(\mathbf{x})|
$$

$$
= \mathcal{L}_{\mathcal{D}'}(f) + L \underset{\boldsymbol{\omega}}{\mathbb{E}} \underset{\mathbf{x}}{\mathbb{E}} \, |f(\mathbf{x}) - f_{\boldsymbol{\omega}}(\mathbf{x})|
$$

$$
\overset{L^1 \leq L^2}{\leq} \mathcal{L}_{\mathcal{D}'}(f) + L \underset{\boldsymbol{\omega}}{\mathbb{E}} \sqrt{\underset{\mathbf{x}}{\mathbb{E}} \, |f(\mathbf{x}) - f_{\boldsymbol{\omega}}(\mathbf{x})|^2}
$$

$$
\overset{\text{first part of the lemma}}{\leq} \mathcal{L}_{\mathcal{D}'}(f) + \frac{LCR\|f\|_k}{\sqrt{qd}}
$$

381 $\qquad\qquad\qquad\qquad\qquad\qquad\qquad\qquad\qquad\qquad\qquad\qquad\qquad\qquad\qquad\qquad\qquad\qquad\qquad\qquad\qquad\quad$ $\square$

382 We next consider an algorithm for learning $\mathcal{H}_k$, by running SGD on top of random features.

---

**Algorithm 2** SGD on RFS

**Input:** RFS $\psi : \Omega \times \mathcal{X} \to \mathbb{R}^d$, number of random features $q$, loss $\ell$, learning rate $\eta > 0$, batch size $b$, number of steps $T > 0$, access to samples from a distribution $\mathcal{D}$
Sample $\boldsymbol{\omega} \sim \mu^q$
Initialize $\mathbf{v}^1 = 0 \in \mathbb{R}^{q \times d}$
**for** $t = 1, \ldots, T$ **do**
$\quad$ Obtain a mini-batch $S_t = \{(\mathbf{x}_i^t, y_i^t)\}_{i=1}^b \sim \mathcal{D}^b$
$\quad$ Update $\mathbf{v}_{t+1} = \mathbf{v}_t - \eta \nabla \mathcal{L}_{S_t}(\mathbf{v}_t)$ where $\mathcal{L}_{S_t}(\mathbf{v}) = \mathcal{L}_{S_t}(f_{\Psi_{\boldsymbol{\omega}}, \mathbf{v}})$.
**end for**
Choose $t \in [T]$ uniformly at random and return $f_{\Psi_{\boldsymbol{\omega}}, \mathbf{v}_t}$

---

383 **Theorem 10.** *Assume that $\psi$ is factorized and $C$-bounded RFS for $k$, that $\ell$ is convex and $L$-Lipschitz,*
384 *and that $\mathcal{D}$ has $R$-bounded marginal. Let $f$ be the function returned by algorithm 2. Fix a function*

$f^* \in \mathcal{H}_k$. *Then*

$$\mathbb{E}\,\mathcal{L}_\mathcal{D}(f) \leq \mathcal{L}_\mathcal{D}(f^*) + \frac{LRC\|f^*\|_k}{\sqrt{qd}} + \frac{\|f^*\|_k^2}{2\eta T} + \frac{\eta L^2 C^2}{2}$$

*In particular, if* $\|f^*\|_k \leq M$ *and* $\eta = \frac{M}{\sqrt{T}LC}$ *we have*

$$\mathbb{E}\,\mathcal{L}_\mathcal{D}(f) \leq L_\mathcal{D}(f^*) + \frac{LRCM}{\sqrt{qd}} + \frac{LCM}{\sqrt{T}}$$

*Proof.* Denote by $\mathbf{v}^* \in \mathbb{R}^{dq}$ the vector

$$v_i^* = \frac{1}{\sqrt{q}}\left(\check{f}^*(\omega_1), \dots, \check{f}^*(\omega_1)\right)$$

By standard results on SGD (e.g. [22]) we have that given $\omega$,

$$\mathcal{L}_\mathcal{D}(f) \quad \leq \quad \mathcal{L}_\mathcal{D}(f_{\boldsymbol\omega}^*) + \frac{1}{2\eta T}\|\mathbf{v}^*\|^2 + \frac{\eta L^2 C^2}{2}$$

Taking expectation over the choice of $\boldsymbol\omega$ and using theorem 4 and equation (5) we have

$$\mathcal{L}_\mathcal{D}(f) \quad \leq \quad \mathcal{L}_\mathcal{D}(f^*) + \frac{LRC\|f^*\|_k}{\sqrt{qd}} + \frac{\|f^*\|_k^2}{2\eta T} + \frac{\eta L^2 C^2}{2}$$

$\square$

We conclude the section with a few calculations of $\check{f}$, that will be useful later.

**Example 11.** Fix $\sigma : \mathbb{R} \to \mathbb{R}$ with Hermite expansion $\sigma = \sum_{n=0}^\infty a_n h_n$ and let $\Omega = \mathbb{R}^d$ and $\mathcal{X} = \mathbb{S}^{d-1}$

1. Consider the RFS $\psi(\omega, \mathbf{x}) = \sigma\left(\langle\omega, \mathbf{x}\rangle\right)$ with $\mu$ being the standard Gaussian measure on $\mathbb{R}^d$. We have that $\psi$ is an RFS for the kernel $k(\mathbf{x}, \mathbf{y}) = \hat{\sigma}\left(\langle\mathbf{x}, \mathbf{y}\rangle\right)$. Consider the function $f(\mathbf{x}) = \langle\mathbf{x}_0, \mathbf{x}\rangle^n$. We claim that $\check{f}(\boldsymbol\omega) = \frac{1}{a_n}h_n\left(\langle\mathbf{x}_0, \omega\rangle\right)$. Indeed, we have,

$$
\begin{aligned}
\mathop{\mathbb{E}}_{\omega\sim\mu}\sigma\left(\langle\omega, \mathbf{x}\rangle\right)\frac{1}{a_n}h_n\left(\langle\mathbf{x}_0, \omega\rangle\right) &= \frac{1}{a_n}\sum_{k=0}^\infty \mathop{\mathbb{E}}_{\omega\sim\mu} a_k h_k\left(\langle\omega, \mathbf{x}\rangle\right)h_n\left(\langle\mathbf{x}_0, \boldsymbol\omega\rangle\right) \\
&= \frac{1}{a_n}\sum_{k=0}^\infty a_k \delta_{kn}\langle\mathbf{x}, \mathbf{x}_0\rangle^k \\
&= \langle\mathbf{x}, \mathbf{x}_0\rangle^n
\end{aligned}
$$

and

$$\left\|\omega \mapsto \frac{1}{a_n}h_n\left(\langle\mathbf{x}_0, \omega\rangle\right)\right\|_{L^2(\Omega)} = \mathop{\mathbb{E}}_{\omega\sim\mu}\frac{1}{a_n^2}h_n^2\left(\langle\mathbf{x}_0, \omega\rangle\right) = \frac{1}{a_n^2} = \|f\|_k^2$$

2. Consider the NTK RFS $\psi(\omega, \mathbf{x}) = \sigma\left(\langle\omega, \mathbf{x}\rangle\right)\mathbf{x}$ with $\mu$ being the standard Gaussian measure on $\mathbb{R}^d$. We have that $\psi$ is an RFS for the kernel $k(\mathbf{x}, \mathbf{y}) = \langle\mathbf{x}, \mathbf{y}\rangle\hat{\sigma}\left(\langle\mathbf{x}, \mathbf{y}\rangle\right)$. Consider the function $f(\mathbf{x}) = \left(\langle\mathbf{x}_0, \mathbf{x}\rangle\right)^n$. As in the item above, it is not hard to show that $\check{f}(\omega) = \frac{1}{a_{n-1}}h_{n-1}\left(\langle\mathbf{x}_0, \omega\rangle\right)\mathbf{x}_0$.

## A.3 Reduction of NN learning to SGD over vector random features

We will prove our result via a reduction to linear learning over the initial neural tangent kernel space, corresponding the the hidden weights.

That is, we define by $\Psi_\mathbf{W}(\mathbf{x})$ the gradient of the function $\mathbf{W} \mapsto h_\mathbf{W}(\mathbf{x})$ w.r.t. the hidden weights. Namely,

$$\Psi_\mathbf{W}(\mathbf{x}) = (u_1\sigma'(\langle\mathbf{w}_1, \mathbf{x}\rangle)\mathbf{x}, \dots, u_{2q}\sigma'(\langle\mathbf{w}_{2q}, \mathbf{x}\rangle)\mathbf{x}) \in \mathbb{R}^{2q\times d}$$

Denote $f_{\Psi_\mathbf{W}, \mathbf{V}}(\mathbf{x}) = \langle\mathbf{V}, \Psi_\mathbf{W}(\mathbf{x})\rangle$ and consider algorithm 3.

It is not hard to show that by taking large enough $B$, algorithm 1 is essentially equivalent to algorithm 3. Namely,

**Algorithm 3** Neural Tangent Kernel Training

---

**Input:** Network parameters $\sigma$ and $d, q$, loss $\ell$, learning rate $\eta > 0$, batch size $b$, number of steps $T > 0$, access to samples from a distribution $\mathcal{D}$
Sample $\mathbf{W} \sim \mathcal{I}(d, q, 1)$
Initialize $\mathbf{V}^1 = 0 \in \mathbb{R}^{2q \times d}$
**for** $t = 1, \dots, T$ **do**
    Obtain a mini-batch $S_t = \{(\mathbf{x}_i^t, y_i^t)\}_{i=1}^b \sim \mathcal{D}^b$
    Using back-propagation, calculate the gradient $\nabla$ of $\mathcal{L}_{S_t}(\mathbf{V}) = \mathcal{L}_{S_t}(f_{\Psi_{\mathbf{W}}, \mathbf{V}})$ at $\mathbf{V}^t$
    Update $\mathbf{V}^{t+1} = \mathbf{V}^t - \eta \nabla$
**end for**
Choose $t \in [T]$ uniformly at random and return $f_{\Psi_W, \mathbf{V}_t}$

---

**Lemma 12.** *Fix a decent activation $\sigma$ as well as convex a decent loss $\ell$. There is a choice $B = poly(d, q, 1/\eta, T, 1/\epsilon)$, such that for every input distribution the following holds. Let $h_1, h_2$ be the functions returned algorithm 1 with parameters $d, q, \frac{\eta}{B^2}, b, B, T$ and algorithm 3 with parameters $d, q, \eta, b, T$. Then, $|\mathbb{E}\,\mathcal{L}_{\mathcal{D}}(h_1) - \mathbb{E}\,\mathcal{L}_{\mathcal{D}}(h_2)| < \epsilon$*

*Proof.* (sketch) For simplicity, instead of assuming that $\sigma$ is $M$-decent, we assume that the activation is twice differentiable and satisfies $\|\sigma'\|_\infty, \|\sigma''\|_\infty < M$. At the end of the sketch we will later explain how to handle $M$-decent activation.

Consider a run of algorithm 1 starting from the initial weights $\mathbf{W} = (W, \mathbf{u})$ in the support of $\mathcal{I}(d, q, 1)$. Consider now another run, running on the same mini-batches, hyper-parameters and initial weights, except that in the second run the output weight are multiplied by $B$, and the learning rate is multiplied by $\frac{1}{B^2}$. Our goal is to show that for large $B$, the second run approximates algorithm 3, with the approximation becoming better as $B$ gets larger.

The process of multiplying the output weights by $B$ cause the gradient, $\nabla_W h_{\mathbf{W}}(x)$, of the hidden layer to be multiplied by $B$, and the gradient, $\nabla_{\mathbf{u}} h_{\mathbf{W}}(x)$, of the output layer to remain the same. Thus, for large enough $B$, we can use this observation in order to ignore the gradient of the output weights. We therefore assume that algorithm 1 only updates the hidden weight. Likewise, while the gradient is multiplied by $B$, the step size is multiplied by $\frac{1}{B^2}$. Hence, the total movement is multiplied by $\frac{1}{B}$. It therefore holds that the optimization process takes place in a ball of radius $\frac{R}{B}$ around $W$, where $R = poly(M, d, q, 1/\eta, T, 1/\epsilon)$ does not depend on $B$.

Now by multiplying the output weights by $B$, we move from the network function $h_W(x)$ to $\tilde{h}_W(x) := B h_W(x)$. The first order approximation of $\tilde{h}$ around the initial weights is

$$\tilde{h}_{W+V}(x) = B h_W(x) + B \langle \nabla_W h_W(x), V \rangle + \frac{H}{2}\|V\|^2 = B \langle \nabla_W h_W(x), V \rangle + \frac{H}{2}\|V\|^2$$

Where $H$ is a uniform bound on the Hessian of $h_W(x)$ (such a bound exists since $\|\sigma'\|_\infty, \|\sigma''\|_\infty < M$). Now, since the optimization in a ball of radius $\frac{R}{B}$ around $W$, we can ignore the quadratic part for large enough $B$, and reduce to the case of optimization over the linear function $B \langle \nabla_W h_W(x), V \rangle$ with learning rate of $\frac{\eta}{B^2}$ starting at $0$. This is equivalent to optimization over the linear function $\langle \nabla_W h(W, x), V \rangle$ with learning rate of $\eta$ starting at $0$, which is exactly algorithm 3.

Finally, to handle general $M$-decent activation, we note that any such activation locally satisfies, $\|\sigma'\|_\infty, \|\sigma''\|_\infty < M$. Now, for large enough $B$, the output of the hidden layer, before the activation, barely moves throughout the optimization process, and hence, for each example in the min-batches, we don't move between different regions in which $\sigma$ satisfies $\|\sigma'\|_\infty, \|\sigma''\|_\infty < M$.

$\square$

By lemma 11 in order to prove theorem 5 it is enough to analyze algorithm 3. Specifically, theorem 5 follows form the following theorem:

**Theorem 13.** *Given $d$, $M > 0$, $R > 0$ and $\epsilon > 0$ there is a choice of $q = \tilde{O}\left(\frac{M^2 R^2}{d\epsilon^2}\right)$, $T = O\left(\frac{M^2}{\epsilon^2}\right)$, as well as $\eta > 0$, such that for every $R$-bounded distribution $\mathcal{D}$ and batch size $b$, the function $h$ returned by algorithm 3 satisfies $\mathbb{E}\,\mathcal{L}_{\mathcal{D}}(h) \leq \mathcal{L}_{\mathcal{D}}\left(\mathcal{H}_{\mathrm{tk}_\sigma^h}^M\right) + \epsilon$*

446  Our next step is to rephrase algorithm 3 in the language of (vector) random features. We note that
447  algorithm 3 is SGD on top of the random embedding $\Psi_{\mathbf{W}}$. This embedding composed of $q$ i.i.d.
448  random mappings $\psi_{\mathbf{w}}(\mathbf{x}) = (\sigma'(\langle \mathbf{w}, \mathbf{x} \rangle)\mathbf{x}, -\sigma'(\langle \mathbf{w}, \mathbf{x} \rangle)\mathbf{x})$ where $\mathbf{w} \in \mathbb{R}^d$ is a standard Gaussian.
449  This can be slightly simplified to SGD on top of the i.i.d. random mappings $\psi_{\mathbf{w}}(\mathbf{x}) = \sigma'(\langle \mathbf{w}, \mathbf{x} \rangle)\mathbf{x}$.
450  Indeed, if we make this change the inner products between the different examples, after the mapping
451  is applied, do not change (up to multiplication by $\sqrt{2}$), and SGD only depends on these inner products.
452  This falls in the framework of learning with (vector) random features scheme, which we define next,
453  and analyze in the next section.

454  We note that since the NTK RFS is factorized and $C$-bounded (for $C = \|\sigma'\|_\infty$), theorem 12 follows
455  from theorem 10. Together with lemma 11, this implies theorem 5.

## A.4  Memorization of random set of points – proof of theorem 7

457  Consider the NTK RFS $\psi(\omega, \mathbf{x}) = \sigma'(\langle \omega, \mathbf{x} \rangle) \mathbf{x}$ with $\mu$ being the standard Gaussian measure on $\mathbb{R}^d$.
458  Recall that $\psi$ is an RFS for the kernel $\mathrm{tk}_\sigma^h(\mathbf{x}, \mathbf{y}) = \langle \mathbf{x}, \mathbf{y} \rangle \, \hat{\sigma}'(\langle \mathbf{x}, \mathbf{y} \rangle)$. As in the proof of theorem
459  5, it is enough to show that for $q = \tilde{O}\left(\frac{m}{d}\right) = \tilde{O}\left(d^{c-1}\right)$, w.p. $1 - o(1)$ over the choice of $S$ and
460  $\boldsymbol{\omega} = (\omega_1, \ldots, \omega_q)$, there is $\mathbf{v} \in \mathbb{R}^{dq}$ such that

$$\langle \mathbf{v}, \Psi_{\boldsymbol{\omega}}(\mathbf{x}_i) \rangle = y_i + o(1) \text{ for all } i \text{ and } \|\mathbf{v}\|_2^2 = \tilde{O}(m) \tag{7}$$

461  Choose a constant integer $c' > 4c + 2$ such that $a_{c'-1} \neq 0$. Such a constant exists since $\sigma$ is not a
462  polynomial. Define

$$f(\mathbf{x}) = \sum_{i=1}^{m} y_i \left(\langle \mathbf{x}_i, \mathbf{x} \rangle\right)^{c'}$$

463

464  **Lemma 14.** *With probability $1 - \delta$ we have that*

$$f(\mathbf{x}_i) = y_i + O\left(\frac{\log^{\frac{c'}{2}}(d/\delta)}{d}\right) \text{ for all } i \text{ and } \|f\|_{k_\sigma}^2 = O(m) + O\left(\frac{\log^{\frac{c'}{2}}(d/\delta)}{d}\right)$$

465  *Proof.* W.p $1 - \delta$ we have that $\langle \mathbf{x}_i, \mathbf{x}_j \rangle \leq O\left(\sqrt{\frac{\log(m/\delta)}{d}}\right) = O\left(\sqrt{\frac{\log(d/\delta)}{d}}\right)$ for all $i, j \in [m]$. In
466  this case we have that for any $i$

$$f(\mathbf{x}_i) = y_i + O\left(m\left(\frac{\log(d/\delta)}{d}\right)^{\frac{c'}{2}}\right) = y_i + O\left(\log^{\frac{c'}{2}}(d/\delta) \, d^{c-\frac{c'}{2}}\right) = y_i + O\left(\frac{\log^{\frac{c'}{2}}(d/\delta)}{d}\right)$$

467  Likewise,

$$\|f\|_{k_\sigma}^2 = a_{c'}^{-2} m + O\left(m^2\left(\frac{\log(d/\delta)}{d}\right)^{\frac{c'}{2}}\right) = a_{c'}^{-2} m + O\left(\log^{\frac{c'}{2}}(d/\delta) \, d^{2c-\frac{c'}{2}}\right) = a_{c'}^{-2} m + O\left(\frac{\log^{\frac{c'}{2}}(d/\delta)}{d}\right)$$

468  $\square$

469  Based on lemma 13, in order to find $\mathbf{v}$ that satisfies equation (7) it is natural to take

$$\mathbf{v} = \frac{1}{\sqrt{q}} \left(\check{f}(\omega_1), \ldots, \check{f}(\omega_q)\right)$$

470  In which case $\mathbb{E}\|\mathbf{v}\|_2^2 = \|f\|_{k_\sigma}^2$ and $\mathbb{E}\left[\langle \mathbf{v}, \Psi_{\boldsymbol{\omega}}(\mathbf{x}) \rangle\right] = \mathbb{E}\left[f_{\boldsymbol{\omega}}(\mathbf{x})\right] = f(\mathbf{x})$. In fact, theorem 4 together
471  with Chebyshev's inequality indeed implies that for large $q$ equation (7) holds. However, this analysis
472  requires $q \approx \frac{m^2}{d}$ while we want $q \approx \frac{m}{d}$. In the remaining part of this section we undertake a more
473  delicate anlysis of the rate in which $f_{\boldsymbol{\omega}}$ approximates $f$ in our specific case. This analysis will imply
474  that $q = \tilde{O}\left(\frac{m}{d}\right)$ suffices for equation (7) to hold w.h.p. Indeed, we will prove that

**Lemma 15.** *W.p.* $1 - \delta - 2^{\Omega(d)}$ *over the choice of $S$ and $\boldsymbol{\omega}$, we have that*

$$\forall i \in [m], \quad |f_{\boldsymbol{\omega}}(\mathbf{x}_i) - f(\mathbf{x}_i)| \leq O\left(\sqrt{\frac{m \log^{c'+2}(m/\delta)}{dq}}\right)$$

Togeter with lemma 13 and Markov's inequality we have

**Theorem 16.** *W.p.* $1 - \delta - 2^{\Omega(d)}$ *over the choice of $S$ and $\boldsymbol{\omega}$, we have that*

$$\langle \mathbf{v}, \Psi_{\boldsymbol{\omega}}(\mathbf{x}_i) \rangle = f_{\boldsymbol{\omega}}(\mathbf{x}_i) = y_i + O\left(\frac{\log^{\frac{c'}{2}}(d/\delta)}{d}\right) + O\left(\sqrt{\frac{d^{c-1} \log^{c'+2}(d/\delta)}{q}}\right) \text{ for all } i$$

*and*

$$\|\mathbf{v}\|_2^2 = O(m/\delta) + O\left(\frac{\log^{\frac{c'}{2}}(d/\delta)}{d\delta}\right)$$

Choosing $\delta = \frac{1}{\log(m)}$ we get that for $q = \tilde{O}\left(d^{c-1}\right)$ equation (7) holds w.p. $1 - o(1)$. This proves theorem 7. The remaining part of the section is a proof of lemma 14. We will need the following version of Hoeffding's bound. A distribution $\mu$ on $\mathbb{R}$ is called $(\delta, B)$-*bounded* if $\Pr_{X \sim \mu}(|X| > B) \leq \delta$.

**Lemma 17.** *Let $\mu$ be a $(\delta, B)$-bounded distribution and let $X_1, \ldots, X_m$ be i.i.d. r.v. from $\mu$. Then, w.p.* $1 - m\delta - \delta'$

$$\left| \mathbb{E}_{X \sim \mu}[X] - \frac{1}{m} \sum_{i=1}^{m} X_i \right| \leq B\sqrt{\frac{2 \ln(\delta'/2)}{m}} + \frac{2\sqrt{\delta \mathbb{E}_{X \sim \mu} X^2}}{1 - \delta}$$

*Proof.* We note that given that $X_i \in [-B, B]$ for all $i$ we have by Hoeffding's bound that w.p. $1 - \delta'$

$$\left| \frac{1}{m} \sum_{i=1}^{m} X_i - \mathbb{E}_{X \sim \mu}[X | X \in [-B, B]] \right| \leq B\sqrt{\frac{2 \ln(\delta'/2)}{m}}$$

We note that

$$
\begin{aligned}
\mathbb{E}_{X \sim \mu}[X | X \in [-B, B]] &= \frac{\mathbb{E}_{X \sim \mu} X + \delta \mathbb{E}_{X \sim \mu}[X | X \notin [-B, B]]}{1 - \delta} \\
&= \frac{\mathbb{E}_{X \sim \mu} X + \mathbb{E}_{X \sim \mu}[X \mathbb{1}[X \notin [-B, B]]]}{1 - \delta}
\end{aligned}
$$

Hence, by Cauchy-Schwartz,

$$\left| \mathbb{E}_{X \sim \mu}[X | X \in [-B, B]] - \mathbb{E}_{X \sim \mu}[X] \right| \leq \frac{\delta}{1 - \delta} \left| \mathbb{E}_{X \sim \mu} X \right| + \frac{\sqrt{\delta \mathbb{E}_{X \sim \mu} X^2}}{1 - \delta} \leq \frac{2\sqrt{\delta \mathbb{E}_{X \sim \mu} X^2}}{1 - \delta}$$

$\square$

Recall now that by example **??**

$$\check{f}(\omega) = \sum_{i=1}^{m} \frac{y_i}{a_{c'-1}} h_{c'-1}(\langle \mathbf{x}_i, \omega \rangle) \mathbf{x}_i$$

Hence, for any $\mathbf{x}$,

$$f_{\boldsymbol{\omega}}(\mathbf{x}) = \frac{1}{q} \sum_{j=1}^{q} \sum_{i=1}^{m} \frac{y_i}{a_{c'-1}} h_{c'-1}(\langle \mathbf{x}_i, \omega_j \rangle) \langle \mathbf{x}_i, \mathbf{x} \rangle \sigma(\langle \omega_j, \mathbf{x} \rangle)$$

In particular, fixing $S$, $f_{\boldsymbol{\omega}}(\mathbf{x})$ is an average of the $q$ i.i.d. random variables

$$f_{\boldsymbol{\omega}}(\mathbf{x}) = \frac{1}{q} \sum_{j=1}^{q} Y(\omega_i, \mathbf{x})$$

Where

$$Y(\omega, \mathbf{x}) = \sum_{i=1}^{m} \frac{y_i}{a_{c'-1}} h_{c'-1}(\langle \mathbf{x}_i, \omega \rangle) \langle \mathbf{x}_i, \mathbf{x} \rangle \sigma(\langle \omega, \mathbf{x} \rangle)$$

**Lemma 18.** *W.p.* $\geq 1 - \delta$ *over the choice of $S$, we have that for every $i \in [m]$, $Y(\omega, \mathbf{x}_i)$ is* $\left(\delta + 2^{-\Omega(d)}, O\left(\sqrt{\frac{m \log^{c'+1}(m/\delta)}{d}}\right)\right)$*-bounded.*

*Proof.* Fix $\omega$ with $\|\omega\| \leq 2\sqrt{d}$. We have that $Y(\omega, \mathbf{x}_i)$, as a function of $S$, is a random variable that is a sum of a single random variable (the summand that corresponds $\mathbf{x}_i$) that is $\left(\delta, O\left(\sqrt{\log^{c'-1}(1/\delta)}\right)\right)$-bounded, as well as $(m-1)$ additional i.i.d random variables that have mean 0, are $\left(\delta, O\left(\sqrt{\frac{\log^{c'}(1/\delta)}{d}}\right)\right)$-bounded, and has second moment $O\left(\frac{1}{d}\right)$. By lemma 16 we have that

$$|Y(\omega, \mathbf{x}_i)| \leq O\left(\sqrt{\frac{m \log^{c'+1}(1/\delta)}{d}}\right) + O\left(\frac{2m\sqrt{\delta/d}}{1-\delta}\right)$$

w.p. $1 - (m+1)\delta$. Equivalently,

$$|Y(\omega, \mathbf{x}_i)| \leq O\left(\sqrt{\frac{m \log^{c'+1}(m/\delta)}{d}}\right) + O\left(\frac{2\sqrt{(m+1)\delta/d}}{1-\delta}\right) = O\left(\sqrt{\frac{m \log^{c'+1}(m/\delta)}{d}}\right)$$

w.p. $1 - \delta$. We have shown that

$$\mathbb{E}_\omega \mathbb{E}_S \left[ 1 \left[ |Y(\omega, \mathbf{x}_i)| \geq O\left(\sqrt{\frac{m \log^{c'+1}(m/\delta)}{d}}\right) \text{ and } \|\omega\| \leq 2\sqrt{d} \right] \right] \leq \delta$$

Changing the order of summation and using Markov, we get that w.p. $\geq 1 - \sqrt{\delta}$ over the choice of $S$, we have that

$$\Pr_\omega \left[ |Y(\omega, \mathbf{x}_i)| \geq O\left(\sqrt{\frac{m \log^{c'+1}(m/\delta)}{d}}\right) \text{ and } \|\omega\| \leq 2\sqrt{d} \right] \leq \sqrt{\delta}$$

Replacing $\delta$ with $\sqrt{\delta}$ and using the fact that $\log\left(m/\delta^2\right) \leq 2\log(m/\delta)$ we get that that w.p. $\geq 1 - \delta$ over the choice of $S$, we have that

$$\Pr_\omega \left[ |Y(\omega, \mathbf{x}_i)| \geq O\left(\sqrt{\frac{m \log^{c'+1}(m/\delta)}{d}}\right) \text{ and } \|\omega\| \leq 2\sqrt{d} \right] \leq \delta$$

Hence, since $\Pr_\omega\left(\|\omega\| > 2\sqrt{d}\right) \leq 2^{-\Omega(d)}$, we conclude that w.p. $\geq 1 - \delta$ over the choice of $S$, $Y(\omega, \mathbf{x}_i)$ is $\left(\delta + 2^{-\Omega(d)}, O\left(\sqrt{\frac{m \log^{c'+1}(m/\delta)}{d}}\right)\right)$-bounded. Finally, using a union bound, and the fact that $\log\left(m^2/\delta\right) \leq 2\log(m/\delta)$ we conclude that w.p. $\geq 1 - \delta$ over the choice of $S$, we have that for every $i \in [m]$, $Y(\omega, \mathbf{x}_i)$ is $\left(\delta + 2^{-\Omega(d)}, O\left(\sqrt{\frac{m \log^{c'+1}(m/\delta)}{d}}\right)\right)$-bounded. $\square$

*Proof.* (of lemma 14) By lemma 17 we conclude that w.p $1 - \delta$ over the choice of $S$, for every $i$, $f_{\boldsymbol{\omega}}(x_i)$ is an average of $q$ i.i.d. $\left(\delta + 2^{-\Omega(d)}, O\left(\sqrt{\frac{m \log^{c'+1}(m/\delta)}{d}}\right)\right)$-bounded random variables. Furthermore, the second moment of each of these variables is $O(m)$. Using lemma 16 we have that w.p. $1 - (m+1)\delta - m2^{-\Omega(d)}$ over the choice of $\boldsymbol{\omega}$,

$$|f_{\boldsymbol{\omega}}(\mathbf{x}_i) - f(\mathbf{x}_i)| \leq O\left(\sqrt{\frac{m \log^{c'+2}(m/\delta)}{dq}}\right)$$

514 Using the assumption that $m = d^c$ and simple manipulation we get that w.p. $1 - \delta - 2^{-\Omega(d)}$ over the
515 choice of $\boldsymbol{\omega}$,

$$|f_{\boldsymbol{\omega}}(\mathbf{x}_i) - f(\mathbf{x}_i)| \le O\left(\sqrt{\frac{m \log^{c'+2}(m/\delta)}{dq}}\right)$$

516 $\qquad\qquad\qquad\qquad\qquad\qquad\qquad\qquad\qquad\qquad\qquad\qquad\qquad\qquad\qquad\qquad\qquad$ □

## A.5 Boundness of distributions

518 Recall that a distribution $\mathcal{D}$ on $\mathbb{S}^{d-1}$ is $R$-bounded if for every $\mathbf{u} \in \mathbb{S}^{d-1}$, $\mathbb{E}_{\mathbf{x}\sim\mathcal{D}} \langle \mathbf{u}, \mathbf{x} \rangle^2 \le \frac{R^2}{d}$. We
519 next describe a few examples of 1-bounded and $(1 + o(1))$-bounded distributions.

520 1. The uniform distribution is 1-bounded. Indeed, for any $\mathbf{u} \in \mathbb{S}^{d-1}$ and uniform $\mathbf{x}$ in $\mathbb{S}^{d-1}$
521 we have

$$\mathbb{E}_{\mathbf{x}} \langle \mathbf{u}, \mathbf{x} \rangle^2 = \sum_{i,j} \mathbb{E}_{\mathbf{x}} u_i u_j x_i x_j = \sum_i \mathbb{E}_{\mathbf{x}} u_i^2 x_i^2 = \sum_i u_i^2 \mathbb{E}_{\mathbf{x}} x_i^2 = \frac{1}{d} \sum_i u_i^2 = \frac{\|\mathbf{u}\|^2}{d} = \frac{1}{d}$$

522 2. Similarly, the uniform distribution on the discrete cube $\left\{ -\frac{1}{\sqrt{d}}, \frac{1}{\sqrt{d}} \right\}^d$ is 1-bounded. Indeed,
523 for any $\mathbf{u} \in \mathbb{S}^{d-1}$ and uniform $\mathbf{x}$ in $\left\{ -\frac{1}{\sqrt{d}}, \frac{1}{\sqrt{d}} \right\}^d$ we have

$$\mathbb{E}_{\mathbf{x}} \langle \mathbf{u}, \mathbf{x} \rangle^2 = \sum_{i,j} \mathbb{E}_{\mathbf{x}} u_i u_j x_i x_j = \sum_i \mathbb{E}_{\mathbf{x}} u_i^2 x_i^2 = \sum_i u_i^2 \mathbb{E}_{\mathbf{x}} x_i^2 = \frac{1}{d} \sum_i u_i^2 = \frac{\|\mathbf{u}\|^2}{d} = \frac{1}{d}$$

524 3. Let $\mathcal{D}$ be the uniform distribution on the points $\mathbf{x}_1, \ldots, \mathbf{x}_m \in \mathbb{S}^{d-1}$. Denote by $X$ the $d \times m$
525 matrix whose $i'$ column is $\frac{\mathbf{x}_i}{\sqrt{m}}$ We have

$$
\begin{aligned}
\max_{\mathbf{u}\in\mathbb{S}^{d-1}} \mathbb{E}_{\mathbf{x}\sim\mathcal{D}} \langle \mathbf{u}, \mathbf{x} \rangle^2 &= \max_{\mathbf{u}\in\mathbb{S}^{d-1}} \frac{1}{m} \sum_{i=1}^m \langle \mathbf{u}, \mathbf{x}_i \rangle^2 \\
&= \max_{\mathbf{u}\in\mathbb{S}^{d-1}} \frac{1}{m} \sum_{i=1}^m \mathbf{u}^T \mathbf{x}_i \mathbf{x}_i^T \mathbf{u} \\
&= \max_{\mathbf{u}\in\mathbb{S}^{d-1}} \mathbf{u}^T X X^T \mathbf{u} \\
&= \|X\|^2
\end{aligned}
$$

526 Hence, $\mathcal{D}$ is $\|X\|$-bounded. In particular, by standard results in random matrices (e.g.
527 theorem 5.39 in [24]), if $\{\mathbf{x}_i\}_{i=1}^m$ are independent and uniform points in the sphere and
528 $m = \omega(d)$ then w.p. $1 - o(1)$ over the choice of the points, $\mathcal{D}$ is $(1 + o(1))$-bounded.

529 4. The uniform distribution on any orthonormal basis $\mathbf{v}_1, \ldots, \mathbf{v}_d$ is 1-bounded. Indeed, for any
530 $\mathbf{u} \in \mathbb{S}^{d-1}$ and uniform $i \in [d]$ we have

$$\mathbb{E}_i \langle \mathbf{u}, \mathbf{v}_i \rangle^2 = \frac{1}{d} \sum_{i=1}^d \langle \mathbf{u}, \mathbf{v}_i \rangle^2 = \frac{\|\mathbf{u}\|^2}{d} = \frac{1}{d}$$