[Reviews · NeurIPS 2020]

Review 1

Summary and Contributions: This paper studies memorization of data by neural networks (as well as polynomial learning), and proves bounds on the number of neurons required. The approach is based on the NTK framework, but introduces a notion of boundedness of the data distribution that when appropriately exploited allows for improvements over prior results. I think the paper should probably be accepted based on the novelty and strength of the results.

Strengths: As of the date of the arXiv posting of the corresponding paper, these results were state of the art for memorization of random data by NN.

Weaknesses: The main potential for improvement of the paper is in presentation. The logical flow of the paper can likely be improved. How exactly the various assumptions enter into the arguments could be presented in a cleaner way. There is a fair amount of redundancy in the first 8 pages, which reduces the space left to explain the key ideas, and I would in general appreciate a more direct presentation of main ideas.

Correctness: I did not read the appendix carefully, but based on the exposition in the first 8 pages and understanding of related literature, I believe that the results are correct.

Clarity: I found the paper to be fairly understandable, but I think the paper could benefit from a better high-level description of the strategy, how the pieces fit together, and what are the key new ideas relative to prior work versus what are ideas that have appeared before. There are also many small typos, grammatical errors, and minor inaccuracies that make the paper harder to read than it would otherwise be. For instance, in line 174, the definition of \|f\|_k does not seem quite complete (although one can infer the correct definition...).

Relation to Prior Work: I think the relation to prior work is mostly satisfactory. As noted above, it would be nice to briefly say what ideas are new and which are not (with attribution). There has also been a fair amount of work on this topic in the last 6-8 months, and it may be appropriate to add references to these.

Reproducibility: Yes

Additional Feedback: The title appears to be incorrect in the submitted PDF.


Review 2

Summary and Contributions: This paper analyzed Neural Tangent Kernel (NTK) which is a widely used technical tool in recent analyses on neural networks. They give a tight analysis on the rate in which NTK converges to its expectations. Then they use this result to show that SGD on two-layer networks can memorize samples, learn polynomials, and learn certain kernel spaces with near optimal network size, sample complexity, and runtime.

Strengths: 1. The contribution of this work is novel and significant. It provides the first near optimal bound of convergence rate of NTK. This result is also used to prove that SGD on two-layer networks can memorize samples, learn polynomials, and learn certain kernel spaces with near optimal network size, sample complexity, and runtime. 2. The result on NTK should be useful for further works. 3. Theorems and lemmas in the paper are correct and well organized.

Weaknesses: The paper assumes bounded distributions or random sampled inputs. One question is: is the data distribution in practice, e.g., ImageNet bounded? Can this question be answered either theoretically or empirically?

Correctness: Theories in the paper are correct.

Clarity: The paper is well written.

Relation to Prior Work: It clearly discussed how this work differs from previous contributions.

Reproducibility: Yes

Additional Feedback:


Review 3

Summary and Contributions: The paper attempts to show that under certain boundedness conditions, the number of parameters required in a simple 2-layer neural network for memorizing sample, learning polynomials, and certain kernel spaces, need not be large unlike what is needed in several prior works. The paper relaxes strong assumptions like linear separability of the data, and quadratic activations made in earlier works to arrive at near optimal network size results. The main technical contribution is an analysis of convergence of the NTK to its convergence.

Strengths: The analysis of concentration of NTK through random feature schemes is interesting and neat and is a strength of the paper. The claims that SGD being able to find a function that is close to the optimal function is another strength. The analysis is able to consider general Lipschitz losses and activations with bounded sub-gradients.

Weaknesses: The results and proofs heavily rely on a strong assumption that the data distribution is over a $d$-dimensional Euclidean sphere. The paper also assumes R-boundedness for $R = O(1)$. In the worst and the most common real-world case, when $R = O(\sqrt{d})$, the result of theorem 4, 5 and 6 loose the gain of dimension factor. However, the neat technical analysis used despite such strong assumptions, overshadows this weakness. Some comments on the writing - - The paper title differs in the main text. There is another paper publicly available with the same name and different from this work. This needs urgent attention. - $\mathcal{N}(I,0)$ is not defined in Equation (1). - $d_1$ is not defined in Equation (2). - Should it be $tk_{\mathbf{W}}$ in line 185? - $\hat\sigma'$ and $\hat\sigma$ aren't defined in the main paper. It should refer to line 369 in Appendix. - Line 221, should it be $2$-nd moment instead of `variance'?

Correctness: Because the setting is simple enough, it would be great to see a toy simulation verifying the dependence of running time and error with $d$ and $q$. For example - an empirical evaluation of Theorem 7 would be great.

Clarity: The paper is well-written and the analysis is neat. There are certain typos and corrections which can be made to make this even better.

Relation to Prior Work: The paper brings forth important recent works which analyze the same problem. There is a very recent work - 'Network size and weights size for memorization with two-layers neural networks - Bubek et al' which also show very similar results on learnability, and it will be great if the authors can comment on it in their final submission.

Reproducibility: Yes

Additional Feedback: The authors answer most of the queries. I have raised the confidence of my score.


Review 4

Summary and Contributions: After the author response: The authors have addressed some of my concerns in their response. I have increased my score. However, I would like to suggest that the authors should revise the paper to improve clarity and rigorousness: The authors should provide full proofs of Theorem 5 and related lemmas/theorems. Around lines 275-279, it would be clearer if the authors could remind the readers that this setting focuses on uniform inputs and hinge loss. Another minor point is that the submission does not have a broader impact section. I understand this paper focuses on theoretical analysis, but the authors may still consider adding a short broader impact section, since this is required by neurips. =========================================================== This paper studies how over-parameterized two-layer networks memorize samples and learn polynomials. The authors show that networks with \tilde O(m) parameters can memorize m points on the unit sphere, and prove that a certain class of polynomials can also be learned by a network with sufficient parameters.

Strengths: Memorization is an important problem and is of interest to the NeurIPS community. The results of this paper are clearly presented, however certain parts of the proofs are not provided in detail.

Weaknesses: One of my concerns is the rigorousness of the paper. A key lemma, namely Lemma 12 in the supplementary material is only given with a proof sketch. Moreover, in the proof sketch, how the authors handle the general M-decent activation functions is discussed very ambiguously. This makes the results for ReLU activation function particularly questionable. The significance and novelty of this paper compared with the existing results are also not fully demonstrated. 1. It is claimed in this paper that a tight analysis is given on the convergence of NTK to its expectations. However, it is not explained in what sense the analysis is tight. The authors did not compare their convergence analysis with existing ones either (for example, the results by Arora et al. (arXiv:1904.11955)). Based on the supplementary material, it seems that this convergence is directly proved based on Hoeffding’s bound. 2. The title points out “no over-parameterization”, but it is not well explained in what sense are the results considered using “no over-parameterization”. Recent results by on Ji and Telgarsky [17] and Chen et al. (arXiv:1911.12360) have shown polylogarithmic dependency. The authors mentioned that [17] requires the data distribution to be realizable with a margin, but it seems that the margin in [17] is not necessarily a constant and therefore [17] also covers the unrealizable case (maybe with stronger width requirements). Moreover, this paper requires that the data inputs are generated from the uniform distribution over the unit sphere, which is a very strong assumption compared with existing results.

Correctness: I did a high-level check and did not find any obvious mistake in the proof. However, Lemma 12 in the supplementary material, which may be crucial to the condition of the number of parameters, is only given with a proof sketch.

Clarity: The paper is well organized and easy to follow. However, there are still typos, and I suggest the authors should further proofread the paper.

Relation to Prior Work: The authors compared their results with some existing ones. However, a more thorough comparison may be necessary to demonstrate the novelty and significance of this paper.

Reproducibility: Yes

Additional Feedback: The title of the submitted paper does not exactly match its title in the CMT system.

[Author Response · NeurIPS 2020]

We thank the reviewers for their efforts and overall positive feedback. Below we address their main concerns.

1. Bounded distributions: We note that *any* whitened distribution is 1-bounded. Hence, $O(1)$ bounded do arise in practice (as whitening is a very popular reprocessing). We will add a comment about that.

2. Tightness of the analysis: The analysis is tight in the sense that the bound in theorem 4 is optimal, up to constant. To the best we know, previews results do not imply that, despite significant efforts in recent years. Moreover, we disagree that the "convergence is directly proved based on Hoeffding's bound". This is far from being true. In order to establish our result we developed a new methodology to analyze vector random features, and used the boundedness of the distribution in a delicate way. The best evidence that the analysis is not trivial is that the result is new despite very significant research in recent years by top researchers – there were more than 20 papers devoted to memorization and NTK, and none of them derived such a convergence result. It is clear that an effort has been made to derive such a result, as the rate in which the NTK converge is central in the analysis of most of these papers.

3. "it is not well explained in what sense are the results considered using "no over-parameterization": No over parametrization means that $\tilde{O}(m)$ parameters are enough to memorize $m$ points. This is standard terminology in related literature. We will add details about this and will make it clearer in the final version.

4. Presentation and details: We will make any effort to improve the writing and add proof details along the lines raised by the reviewers.

[Meta-Review · NeurIPS 2020]

This paper studies optimization in the NTK regime, further improving the best prior width bounds for random data (I believe Oymak-Soltanolkotabi were the prior best). The reviewers and I were all favorable, and I look forward to seeing this paper appear, and support the authors in further investigations. --- Minor comments. (a) A reviewer pointed out that Lemma 12 did not receive a full proof: please provide a complete proof in your revisions. Relatedly, this point was not sufficiently handled in the rebuttal, despite the rebuttal using less than half a page. Please consider such things in the future. (b) I believe there are some newer works now which should be cited? One is by Roman Vershynin, and I believe Sebastien Bubeck and colleagues also had a paper on the "Baum" problem. (c) The title of the paper does not match between tex and CMT?!